# Stealth Luminescent Organic Nanoparticles Made from Quadrupolar Dyes for Two-Photon Bioimaging: Effect of End-Groups and Core

**DOI:** 10.3390/molecules27072230

**Published:** 2022-03-29

**Authors:** Morgane Rosendale, Jonathan Daniel, Frédéric Castet, Paolo Pagano, Jean-Baptiste Verlhac, Mireille Blanchard-Desce

**Affiliations:** Départment Sciences de la Matière et du Rayonnement, Université Bordeaux, CNRS, Bordeaux INP, ISM, UMR 5255, F-33400 Talence, France; morgane.rosendale@u-bordeaux.fr (M.R.); jonathan.daniel@u-bordeaux.fr (J.D.); frederic.castet@u-bordeaux.fr (F.C.); paolo.pagano8008@gmail.com (P.P.); jean-baptiste.verlhac@u-bordeaux.fr (J.-B.V.)

**Keywords:** quadrupolar dyes, 2P absorption, fluorescence, organic nanoparticles, stealth, surface interactions, bioimaging

## Abstract

Molecular-based Fluorescent Organic Nanoparticles (FONs) are versatile light-emitting nano-tools whose properties can be rationally addressed by bottom-up molecular engineering. A challenging property to gain control over is the interaction of the FONs’ surface with biological systems. Indeed, most types of nanoparticles tend to interact with biological membranes. To address this limitation, we recently reported on two-photon (2P) absorbing, red to near infrared (NIR) emitting quadrupolar extended dyes built from a benzothiadiazole core and diphenylamino endgroups that yield spontaneously stealth FONs. In this paper, we expand our understanding of the structure-property relationship between the dye structure and the FONs 2P absorption response, fluorescence and stealthiness by characterizing a dye-related series of FONs. We observe that increasing the strength of the donor end-groups or of the core acceptor in the quadrupolar (D-π-A-π-D) dye structure allows for the tuning of optical properties, notably red-shifting both the emission (from red to NIR) and 2P absorption spectra while inducing a decrease in their fluorescence quantum yield. Thanks to their strong 1P and 2P absorption, all FONs whose median size varies between 11 and 28 nm exhibit giant 1P (10^6^ M^−1^.cm^−1^) and 2P (10^4^ GM) brightness values. Interestingly, all FONs were found to be non-toxic, exhibit stealth behaviour, and show vanishing non-specific interactions with cell membranes. We postulate that the strong hydrophobic character and the rigidity of the FONs building blocks are crucial to controlling the stealth nano-bio interface.

## 1. Introduction

Fluorescent nanoparticles (NPs) are fast becoming indispensable in the biomedical toolkit with applications ranging from drug and gene delivery [1,2] to bioimaging and diagnosis [3,4]. To this aim, they should be capable of absorbing and emitting light above 650 nm—i.e., within the biological transparency window [5]—and be bright—ideally such that single emitters can be detected. All organic fluorescent NPs are exquisitely sensitive to the chemical nature of the chromophores composing them such that these properties can be molecularly engineered. Red-shifted emission of the chromophores can be achieved in two ways: (i) intra-molecularly via the induction of a dipolar moment between electro-donor and electro-acceptor moieties [6,7], or via extended π-conjugation [8,9], or (ii) inter-molecularly by favouring the formation of J-type aggregates [10,11]. Interestingly, push-pull systems containing electron-donating and electron-withdrawing moieties such as dipolar and to an even greater extent, quadrupolar molecules, also display two-photon absorption (2PA) properties [12,13]. This non-linear phenomenon is defined by the simultaneous absorption of two low-energy photons by an excitable compound. The wavelength used for 2P excitation is thus favourably red-shifted as compared to that used for one-photon (1P) excitation. However, extending the π-conjugation in systems as confined as NPs may be deleterious to fluorescence emission due to π-stacking [14]. Obtaining bright, red, organic NPs, therefore, requires a fine understanding of nano-confinement effects.

Another consideration to keep in mind when using NPs in a biological environment is that contrary to small molecules, the size of NPs is in the range of bio-macromolecules (proteins, glycans…). This characteristic results in important interactions with cells. Intense research efforts have been deployed towards understanding the link between the physicochemical nature of an NP’s core and its behaviour towards cellular membranes [15,16,17]. We thus know that size, charge, colloidal stability, hydrophilicity, elasticity, etc., of nano-materials influence their cellular internalisation kinetics. However, these parameters can be shielded by the common practice of NP surface coating. This process is often applied to address NP fouling by circulating biomolecules [18,19]. Arguably, the most common aim of coating is to make NPs stealthier to increase their bioavailability thanks to an increased circulation time [20]. However, even the most popular coating polymer, polyethylene glycol (PEG), raises toxicological concerns such that more biocompatible alternatives need to be discovered before coated NPs can find their way to the clinic [21].

In this paper, we address this issue by investigating an original molecular design that integrates spectral and surface considerations in single-component based NPs. Fluorescent Organic Nanoparticles (FONs) are prepared by nanoprecipitation in water of rationally designed organic dyes [22,23]. Tuning the spectral properties of FONs by molecular engineering is an active and versatile field of studies [24,25,26,27]. However, controlling the fate of FONs in cellular environments has been seldom addressed. We have recently reported on two dye designs, bis-dipolar [28] and quadrupolar [29], that yield spontaneously stealth FONs. We then modulated the π-conjugation of the quadrupolar template of the composing chromophores to yield FONs of red-shifted emission. However, we found that these slight modifications of the chromophores’ chemical structure could influence the stealth behaviour of their derived FONs [30]. Here, we further explore this link between chromophore structure and FONs nano-bio interactions by presenting a structurally related series of chromophores varying in the strength of their dipolar moments. To do so, starting from a dye template of known properties [30], we designed novel chromophores bearing triphenylamine end-groups either weakened or strengthened in their electron-donating character—by substitution with aldehydes or methoxy groups respectively—as well as a chromophore having a core with increased electro-acceptor strength—by introduction of a (bis)benzothiadiazole motif. We report on the photo-physical properties of these dyes, including their 2PA capacity and red to NIR emission, study the effects of nano-confinement on those properties and show that the stealth behaviour of their derived FONs can be maintained and even enhanced in the presence of these modifications. The bulky, hydrophobic, quadrupolar template used in this study thus appears to be key in allowing a nano-organisation of chromophores within and at the surface of FONs that confers them with a spontaneously stealth character.

## 2. Results and Discussion

### 2.1. Design

Molecular-based fluorescent FONs may be obtained from the spontaneous aggregation of suitable hydrophobic dyes in water upon nanoprecipitation [31]. The non-covalent confinement of a large number of dye molecules within FONs may promote interchromophoric interactions that strongly affect their optical properties. The chemical structure of the composing dyes is thus critical to obtaining FONs of the desired properties. The design of **Q_2_**, i.e., the chromophore template from which **Q_1_**, **Q_3_** and **Q_4_** are derived in this study, was rationally elaborated with four considerations in mind. (i) A strong hydrophobic character to allow nanoprecipitation in water; (ii) a quadrupolar symmetry (D-π-A-π-D motif) favourable for 2PA; (iii) a planar, π-conjugated ethynyl bridge for red-shifted emission; and (iv) bulky triphenylamine end-groups and butyl side chains to hinder π-stacking by steric hindrance that would be deleterious to fluorescence emission. We have previously reported on the photo-physical properties, confinement effects, and stealth FONs of this chromophore in the context of modulating the extent of conjugation of the π-bridge [30]. In order to further study the structure-property relationships of this dye, we now report on chromophores designed to vary donor and acceptor strengths. These changes were aimed at modulating the efficiency of the intramolecular charge transfer (ICT) operating in these dyes. Triphenylamine donating end-groups (D) strength was bidirectionally modulated by introduction of either an electron-withdrawing aldehyde moiety (**Q_1_**) or electro-donating methoxy moieties (**Q_2_**). Alternatively, core acceptor (A) strength was increased by introducing a second benzothiadiazole (BTDA) moiety to the core (**Q_4_**). We note that on top of increasing the strength of the core from **Q_2_** to **Q_4_**, the introduction of a double BTDA motif also introduces some flexibility in the molecule, which may influence the fluorescence and surface properties. The chemical structures of these novel quadrupolar push-pull systems are provided in Figure 1.

### 2.2. Synthesis of the Dyes

Our strategy to synthesize dyes **Q_1_**–**Q_4_** relied on a linear approach in three key steps: (i) synthesis of the end-group moiety; (ii) synthesis of the electron withdrawing core; (iii) cross-coupling of the end-groups with the core (Figure 1). The dissymmetrical donor precursors **2a** and **2b** were prepared by the Ullmann coupling of 2,7-diiodo-9,9′-dibutylfluorene **1** with diphenylamine or 4-methoxydiphenylamine. Vilsmeier-Haack formylation on **2a** with a slight excess of POCl_3_ favours the introduction of only one aldehyde group on the diphenylamine moiety to generate the donor precursor **3** with a moderate yield. The dissymmetrical end-groups **2a**-**b** and **3** are then engaged in a double Sonogashira cross-coupling with 4,7-diethynyl-2,1,3-benzothiadiazole [32] to give the corresponding dyes **Q_1_**–**Q_3_** with a moderate to very good yield. Dye **Q_4_**, containing the di-benzothiadiazole core, was obtained by a double Sonogashira cross-coupling reaction of the protected bis-acetylene dibenzothiadiazole **5** (previously obtained by a double Sonogashira cross-coupling of trimethylsilylacetylene with 7,7′-diiodo-4,4′-bi-2,1,3-benzothiadiazole synthesised as in [33]) in the presence of an excess of derivative **2a** using an in situ deprotection of the alkyne by tetrabutylammonium fluoride.

### 2.3. Photophysical Properties of the Quadrupolar Dyes in Solution

#### 2.3.1. Absorption

As illustrated in Figure 3 and Table 1, all compounds show an intense absorption band in the blue-green region, characteristic of an ICT transition, which is confirmed by Density Functional Theory (DFT) calculations that evidence a periphery to core charge redistribution occurring from the donating diphenylamine moieties to the BTDA core upon the S_0_→S_1_ transition (Figure 2). Intense absorption bands located in the ultraviolet spectral region (UV) are also observed (Table 1), which are also well reproduced by DFT calculations (Table 2).

We note that increasing the strength of the electron-releasing end-groups (**Q_1_**→**Q_2_**→**Q_3_**) or the strength of the acceptor core (**Q_2_**→**Q_4_**) induces a broadening (except in the case of **Q_3_**) and a bathochromic shift of the ICT absorption band. Again, DFT calculations reproduce the experimental redshifts well, which come with a slight decrease in the dipole moment variation upon excitation (Table 2).

The two acceptor substituents of **Q_1_** as well as the four donor substituents of **Q_3_** also generate a hyperchromic shift of the ICT band, leading to slightly larger molar absorption coefficients than **Q_2_** (Table 1). This is consistent with the slightly larger oscillator strength values computed for **Q_1_** and **Q_3_**. In contrast, **Q_4_** shows a smaller absorption coefficient than **Q_2_**. As shown by DFT calculations (Figure 2) the presence of the two neighbouring BTDA units in the central core of **Q_4_** induces a deviation from planarity, with a dihedral angle of about 30° between the BTDA planes. However, the oscillator strengths calculated for **Q_2_** and **Q_4_** (Table 2) suggest that the non-planar structure of **Q_4_** is not at the origin of its weaker absorption. Yet, deviations from the optimized value of the dihedral angle around the single bond connecting the BTDA units, possibly induced by thermal effects owing to the expected low rotational barrier, might weaken the conjugation along the molecular backbone and dampen the absorption intensity. Finally, increasing the solvent polarity induces a broadening but does not affect the position of the ICT absorption band much (Figure 3).

#### 2.3.2. Fluorescence

All compounds show intense fluorescence when dissolved in apolar solvents such as toluene (Table 1). Both increasing the strength of the donating end-groups and of the acceptor core induces a broadening of the absorption band, and a bathochromic shift of the emission band with fluorescence shifting from green for **Q_1_** to yellow for **Q_2_** and red for **Q_3_** and **Q_4_** (Figure 3). The polarity of the solvent is found to significantly affect the emission of all four dyes. Indeed, a marked red shift of the emission spectrum and an increase in the Stokes shift values are observed with increasing solvent polarity (Table 1, Figure 3). As a result, the emission is shifted from green in apolar cyclohexane to red (**Q_1_**) or NIR (**Q_2_**–**Q_4_**) in low-medium polarity solvents such as tetrahydrofurane (THF). In parallel, a dramatic decrease in the fluorescence quantum yield is observed (see Table 1) such that all dyes are non-fluorescent in polar solvents and dyes having the strongest donating end-groups (**Q_3_**) or core (**Q_4_**) show vanishing fluorescence even in medium-low polarity solvents.

The marked positive fluorescence solvatochromism indicates that the relaxed emitting state is highly polarised. This points to a breaking of centrosymmetry occurring after excitation, prior to emission leading to a polarised emissive excited state [34,35,36,37]. We note that the Stokes shift values increase with the increasing of solvent polarity (Table 1). In addition, the Stokes shift values were found to be linearly dependent upon the orientation polarizability (Δ*f*) in agreement with the Lippert-Mataga correlation [38,39] (Equation (1)).
(1)νabs−νem=2Δμ2hca3Δf+const
where *ν_abs_* (*ν_em_*) is the wavenumber of the absorption (emission) maximum, *h* is the Planck constant, *c* is the light velocity, *a* is the radius of the Onsager cavity and
(2)Δf=εr−12εr+1−n2−12n2+1
where *ε_r_* is the dielectric constant and *n* the refractive index of the solvent; while Δ*μ* is the change of dipole moment of the solute between the emissive excited state and the corresponding Franck-Condon ground state.

The values of the specific solvatochromic shift (slope values) are reported in Table 1. We note that the polarity of the emissive excited state is found to increase with the strength of electron-releasing end-groups (**Q_1_**→**Q_2_**→**Q_3_**) as well as with the strength of electron-withdrawing core (in particular, taking into account the larger size of the Onsager cavity of **Q_4_** with respect to **Q_2_**). The quenching of fluorescence quantum yield in medium (**Q_3_**, **Q_4_**) to high polarity solvents (**Q_1_**, **Q_2_**) may point to a competitive photo-induced intramolecular electron transfer occurring in the excited state. Such a phenomenon seems even more efficient for dyes **Q_3_** and **Q_4_**, which have the strongest electron-donating end-groups or electron-accepting core (see Table 1).

#### 2.3.3. Two-Photon Absorption Properties

2PA properties were determined by conducting 2P induced fluorescence measurements in solution [40]. Measurements were conducted in toluene as all dyes retain sizeable fluorescence in that solvent. The 2PA spectra are shown in Figure 4. As expected from the quadrupolar nature of the **Q_2_**–**Q_4_** dyes, the lowest energy 1P allowed excited state is almost 2P forbidden, whereas an intense 2PA band is observed at higher energy corresponding to an almost 1P forbidden excited state [12,41,42,43,44,45]. Interestingly, dye **Q_1_** also shows a strong 2PA band located around 750 nm corresponding to a 1P allowed excited state. This specificity may arise from the addition of the aldehyde substituent and may be related to the dipolar contribution of the end-groups. The comparison of the 2PA spectra of the four dyes shows that the presence of the substituents on the electron-donating end-groups does not affect the peak 2PA response much (σ_2_^max^) but definitely affects its spectral range (Figure 4B). The four donating OMe substituents of dye **Q_3_** induce a marked broadening and red-shift of the 2PA spectrum. As a result, **Q_3_** retains large 2PA responses (>500 GM) between 820 nm and 950 nm, whereas **Q_2_** shows significant 2PA only between 800 nm and 900 nm. Oppositely, the two electron-withdrawing substituents (CHO) of dye **Q_1_** generate a blue shift and strong 2PA signal between 720 nm and 850 nm. Finally, we observe that dye **Q_4_** shows a 2–3 times smaller peak 2PA response than dye **Q_2_**. This marked reduction in 2PA capacity, which again may be related to the conformation flexibility of **Q_4_** at the core, is thus even more striking under 2P excitation than under 1P excitation.

### 2.4. Preparation and Characterisation of the FONs

#### 2.4.1. Nanoprecipitation

Nanoprecipitation was easily achieved by fast addition under sonication of a 1 mM stock solution of the dyes in THF into water (1% *v*/*v*). This process almost instantly results in clear, coloured solutions emitting red fluorescence. The colloidal nature of these solutions was confirmed by Transmission Electron Microscopy (TEM). All dyes yielded spherical NPs of small dry diameter (10–20 nm) with **Q_1_** and **Q_3_** FONs also displaying a second population of medium-sized NPs (20–60 nm) (Figure 5, Table 3).

The apparent surface charge of the FONs was determined by measuring their zeta potential. All FONs present a very negative zeta potential (Table 3). Interestingly, **Q_3_** FONs display a measurably less negative surface potential than other FONs, most probably in relation with the positive partial charges born by the methoxy end-groups. Oppositely, **Q_1_** FONs have the most negative potential of the four, which may be attributed to the presence of negative partial charges carried by the aldehyde groups. Finally, the similar potentials of FONs made of **Q_2_** and **Q_4_** suggest that these unsubstituted dyes acquire a similar arrangement at the surface of FONs.

#### 2.4.2. Optical Properties of FONs

FONs made from **Q_1_**–**Q_4_** dyes show intense absorption in the blue-green region. Thanks to the confinement of a large number of dyes, they show giant absorption coefficients (Table 3). A regular bathochromic shift is observed upon increased donating strength of the electron-donating end-groups **Q_1_**→**Q_2_**→**Q_3_** (Figure 6). In contrast, the stronger acceptor core (**Q_2_**→**Q_4_**) does not generate a bathochromic shift of the ICT absorption band.

The confinement of the quadrupolar dyes within FONs leads to a slight hypochromic shift of the ICT band. We note that this effect seems more pronounced in the case of **Q_2_** and **Q_4_** (Figure 7). On the other hand, no sign of excitonic coupling or splitting is observed, in contrast to previous observations on different quadrupolar derivatives built from a fluorene core [11]. This suggests that the four pendant alkyl chains extending above and below the π-conjugated systems hamper such phenomenon.

The fluorescence emission spectra of FONs are blue-shifted with respect to the dyes dissolved in THF (Figure 7) but red shifted compared to in toluene. The Stokes shift values indicate an environment of medium-low equivalent polarity (apparent Δf = 0.15, 0.10, 0.12, 0.11 for **Q_1_**, **Q_2_ Q_3_** and **Q_4_**, respectively)**.** Interestingly, FONs made from dye **Q_1_** having the weaker electron-donating end-groups was found to have a similar emission, even slightly red-shifted, and a larger fluorescence quantum yield than FONs made from dye **Q_2_**. Oppositely, dye **Q_3_** having the strongest electron-donating end-groups shows the most red-shifted emission and the weakest fluorescence quantum yield. The main observation is the marked reduction in fluorescence upon confinement within FONs. In addition, we note that the fluorescence decays are no longer monoexponential. Instead, FONs evidence three lifetime components (Table 1 and Appendix A). Multicomponent decays have been observed previously in FONs made from dipolar (push-pull) or octupolar dyes [46,47]. The shortest lifetime—which is shorter than that of the corresponding molecular dye dissolved in low polarity solvents (except in the case of dye **Q_4_**)—may be tentatively ascribed to dyes located close to the FONs’ surface. Indeed, H-bonded water molecules promote efficient non-radiative deactivation processes via high-energy vibrations. In contrast, the longest lifetime—which is significantly longer than that of the corresponding free dye dissolved in low polarity solvents—may point to the occurrence of specific nano-arrangements of dyes within FONs (i.e., nearly parallel), which would lead to the reduction in radiative decay rates for symmetry reasons. As a whole, the reduction in the fluorescence quantum yield of the dyes induced by their confinement within pure NPs can be mainly ascribed to a large decrease in the radiative rate within FONs as observed from Table 1. In contrast, we note that the non-radiative decay rates are reduced in comparison to those of the dyes dissolved in medium polarity solvents, and similar to those of the dyes in a low polarity solvent (except for dye **Q_2_**). Along this line, we stress that the fluorescence quantum yield of FONs in water (ε_r_ = 80) is significantly larger than that of their corresponding dye in a polar solvent such as acetonitrile (ε_r_ = 31), where all dyes are non-fluorescent. This clearly illustrates that FONs made from quadrupolar dyes can be considered as leading to aggregation caused quenching (ACQ) when compared to dyes dissolved in THF, but to aggregation induced emission (AIE) when compared to dyes dissolved in a polar solvent. Despite the small FONs fluorescence quantum yield values in water, all FONs show large brightness values thanks to the high number of confined dyes per FON and to the large absorption coefficients of the quadrupolar dyes (Table 3). Notably, small FONs (i.e., 11–13 nm) made from dye **Q_1_** show the largest brightness: 1.5 × 10^6^ M^−1^.cm^−1^ compared to 0.7 × 10^6^ M^−1^.cm^−1^ and 0.1 × 10^6^ M^−1^.cm^−1^ for FONs made from dyes **Q_2_** and **Q_3_**, respectively.

2P induced fluorescence measurements conducted on colloidal solutions of FONs made from dyes **Q_1_**–**Q_4_** show that the dyes retain large 2PA responses as FON subunits. The presence of substituents allows for the tuning of the 2PA spectra (Figure 8A): while **Q_1_** FONs show 2PA maxima at 750 nm and 820 nm, **Q_2_** FONs show maxima at 840 nm and 890 nm, and **Q_3_** at 890 nm and 940 nm. Thanks to their broad 2PA bands, **Q_1_** retains large 2PA responses in the 750–900 nm range and **Q_2_** in the 800–940 nm range. As for **Q_3_** FONs, they show significant 2PA response in the whole 800–1000 nm range. Yet, their very low fluorescence quantum yield negatively impacts their brightness as FON subunits. The presence of a stronger acceptor core, on the other hand, has little effect on the location of the 2PA maxima such that **Q_4_** and **Q_2_** FONs display similar peaks. Although a broadening of the response at higher energies can be observed in a similar fashion to **Q_3_**, this effect does not compensate the lower absorption capacity of this dye, such that the 2PA response of **Q_4_** as FON subunits does not exceed that of **Q_2_** in the 900–1000 nm range.

Despite these mixed performances of the newly engineered dyes as FON subunits, it is interesting to consider the brightness of FONs as individual nano-objects containing hundreds to thousands of chromophores for imaging applications. In this context, the population of **Q_1_** and **Q_3_** FONs of a similar size to **Q_2_** FONs (~10 nm) display similar (2 × 10^4^ GM) or lower (4 × 10^3^ GM) 2P brightness values. However, the larger **Q_1_**, **Q_3_** and **Q_4_** populations of FONs (~20–30 nm) display two to ten times exalted brightness values (5 × 10^4^ to 2 × 10^5^ GM, Table 3, Figure 8B). These 2P brightness values are lower than the giant values that have been reported for dye-loaded silica-based NPs (1 × 10^7^ GM [48]). However, the size of these NPs is much larger (~150 nm). When 2P brightness values are considered in relation to NP size [47], our FONs (~10–20 GM/nm^3^) therefore compare favourably to dye-loaded silica NPs (7 GM/nm^3^). On the other hand, the 2P brightness of our FONs is in the same order of magnitude as that of red-emitting water-soluble quantum-dots (5 × 10^4^ GM [49]) which are slightly smaller. The latter, therefore, display a larger 2P brightness relative to their volume (33 GM/nm^3^). Taken together, increasing the strength of the end-groups of our quadrupolar dyes turns out to be less efficient in terms of the ‘2P brightness vs. size’ figure of merit than the modification and lengthening of the π-linker (~100–200 GM/nm^3^ [30]).

### 2.5. Bioimaging: Usability and Interactions with Cells

As described in detail above, the interesting red emission of **Q_2_** FONs is maintained in **Q_1_** FONs and further red-shifted by 65 nm and 30 nm in **Q_3_** and **Q_4_** FONs, respectively. These FONs can, therefore, represent interesting tools for bioimaging. We thus incubated FONs, diluted 100 times in cell culture medium, on monkey fibroblasts (Cos7 cells) for 24 h and assessed their biocompatibility by labelling the cells with Calcein Green-AM (CG) prior to imaging. CG can be used as a viability marker as it becomes fluorescent only upon cleavage by intracellular esterases. We found that after 24 h incubation with FONs, a homogenous green intracellular signal could be observed, attesting to the innocuousness of FONs (Figure 9, leftmost).

As previously described with **Q_2_** FONs [30], we next assessed the stealth behaviour of FONs by z-stack confocal microscopy under 1P or 2P excitation. As a positive control for internalisation, we also imaged FONs previously coated with poly(allylamine hydrochloride) (PAH), a positively charged polymer that promotes NP interactions with cellular membranes [50]. We note that the large Stokes shift of FONs allows for dual colour imaging of CG and FONs with a single excitation wavelength using a 1P 488 nm laser line while collecting green fluorescence around 520 nm and red fluorescence above 600 nm. However, under this configuration, the CG signal leaks into the red channel, diminishing the contrast of the FONs image (data not shown). We thus took advantage of the 2PA capacity of FONs to selectively image them, as CG, a fluorescein derivative, comparatively poorly absorbs around 900 nm under 2P excitation. In agreement with our previous report, we confirmed that bare **Q_2_** FONs are minimally internalised by cells, whereas PAH coated **Q_2_** FONs (FONs@PAH) are efficiently taken up. We further checked whether this minimal uptake could be enhanced by increasing the bare FONs concentration 10 times during the incubation phase. We found that under these conditions, more FONs indeed appeared to have internalised. However, the detected signal did not reach the extent achieved via PAH coating, underlining the slow kinetics of FONs uptake. Of note, increasing the concentration of bare FONs 10 times had no adverse effect on cell viability, as attested by CG imaging (data not shown). We next found that **Q_1_** FONs had a similar behaviour, with minimal cellular uptake when used bare and diluted, increased uptake when incubated at a higher concentration, and efficient uptake following PAH coating. Most interestingly, **Q_3_** and **Q_4_** FONs appeared to be even stealthier than FONs derived from the **Q_2_** template. Indeed, very little signal was detected regardless of the concentration used for FONs incubation. In contrast, FONs@PAH could be well detected intracellularly, indicating that these FONs are bright enough for bioimaging too, when cellular uptake is triggered. We have thus, once again, confirmed that the quadrupolar template of the dye series studied in this report yields FONs of tuneable stealth character.

## 3. Materials and Methods

### 3.1. Chemical Characterisations

Infrared spectra were measured on a Perkin Elmer Spectrum 100 Optica. ^1^H and ^13^C NMR spectra were recorded on a Bruker Advance III 200 spectrometer or Bruker Advance I 300 at 300 MHz and 75 MHz, respectively. Shifts (*δ*) are given in ppm with respect to solvent residual peak and coupling constants (*J*) are given in Hertz. HRMS (FD) were carried out at CESAMO (Bordeaux, France). Melting points were measured on Stuart SMP10.

### 3.2. Quantum Chemical Calculations

Molecular structures were optimized using Density Functional Theory (DFT) in the gas phase with the range-separated CAM-B3LYP exchange-correlation functional [51] in association with the 6-311G(d) Gaussian basis set. Dispersion effects were added by using the Grimme’s D3 correction with Becke-Johnson damping (GD3BJ) [52]. Conformational studies were performed in order to define the torsion angles corresponding to the lowest-energy structure. Vertical transition energies and excited state properties were computed by employing the Time-Dependent Density Functional Theory (TD-DFT) at the CAM-B3LYP/6-311G(d) level. Solvent effects (toluene) were taken into account in the calculations of optical properties by using the non-equilibrium Polarizable Continuum Model (PCM) in its integral equation formalism (IEF) [53]. All calculations were performed using the Gaussian16 package [54].

### 3.3. Dye Synthesis

Commercially available reagents were purchased from Aldrich, TCI, Alfa Aesar or Fluorochem and were used without further purification. Dry solvents were distilled from the appropriate drying reagents [55] immediately before use. All air- or water-sensitive reactions were carried out under argon. Chromatography columns were performed using VWR silica gel Si 60 (40–63 µm, 230–400 mesh).

4,7-diethynyl-2,1,3-benzothiadiazole (**4**) was synthesized according to a slightly modified procedure used from literature [32].

#### 3.3.1. Synthesis of **Q_1_**

9,9-dibutyl-7-iodo-2-diphenylaminofluorene (**2a**): A mixture of 2,7-diiodo-9,9-dibutylfluorene (1.06 g, 2.00 mmol), diphenylamine (360 mg, 2.13 mmol), potassium carbonate (1.90 g, 13.75 mmol), 18-crown-6 (20 mg, 0.07 mmol) and copper powder (90 mg, 1.42 mmol) in o-dichlorobenzene (7 mL) was degassed and heated to reflux for 24 h. When cooled down to room temperature, the suspension was diluted with dichloromethane (50 mL) and filtered on a small Celite pad. The solvents were removed by vacuum distillation and the residue was chromatographed on silica, eluting with a 90/10 petroleum ether/dichloromethane mixture. The fractions containing the desired compound were gathered and concentrated under reduced pressure. Precipitation from petroleum ether afforded a pale grey solid (455 mg, 40%). Mp 129–130 °C. ^1^H NMR (DMSO d6, 300 MHz) δ 7.79 (d, J = 1.3 Hz, 1H), 7.73 (d, J = 8.1 Hz, 1H), 7.66 (dd, J = 7.9; 1.4 Hz, 1H), 7.54 (d, J = 7.9 Hz, 1H), 7.29 (m, 4H), 7.08 (d, J = 2.1 Hz, 1H), 7.04 (m, 6H), 6.94 (dd, J = 8.3; 2.1 Hz, 1H), 1.86 (m, 4H), 1.03 (m, 4H), 0.66 (t, J = 7.5 Hz, 6H), 0.52 (m, 4H). ^13^C NMR (CDCl_3_, 100 MHz) δ 153.26 151.73 148.05 147.91 140.80 135.99 135.23 132.04 129.39 124.16 123.49 122.89 121.02 120.68 119.10 91.69 55.39 40.07 26.17 23.16 14.09.

Benzaldehyde, 4-[(7-iodo-9, 9-dibutyl-9H-fluoren-2-yl)phenylamino] (**3**): Phosphoryl chloride (0.1 mL, 0.4 mmol) was slowly added to a cooled (0 °C) solution of **2a** (215 mg, 0.5 mmol) in dimethylformamide (2.5 mL). The solution was heated at 90 °C for 3 h and then poured on ice and extracted with dichloromethane, washed with sodium bicarbonate, and dried. The oily residue was chromatographed on silica using dichloromethane as the eluent. The fractions containing the desired compound were gathered and concentrated under reduced pressure. The liquid slowly crystallized when cooled and could be recrystallized from methanol with a trace of water (175 mg, 58%). Mp: 120–121 °C. ^1^H NMR (CDCl_3_, 300 MHz) δ 9.87 (s, 1H), 7.42 (d, J = 8.7 Hz, 2H), 7.69 (m, 2H) 7.64 (d, J = 8.1 Hz, 1H), 7.43 (dd, J = 7.3; 1.1 Hz, 1H), 7.38 (m, 2H), 7.21 (m, 3H), 7.15 (m, 2H), 7.12 (d, J= 8.7 Hz, 2H), 1.89 (m, 4H), 1.11 (m, 4H), 0.74 (t, J = 7.4 Hz, 6H), 0.66 (m, 4H). ^13^C NMR (CDCl_3_, 100 MHz) δ 190.46 153.31 153.20 151.99 146.28 145.93 140.14 137.23 136.05 132.05 131.35 129.74 129.34 126.01 125.19 125.09 121.25 120.98 120.77 119.72 92.40 55.39 39.84 26.04 22.96 13.92.

4,7-Bis[(7-N-(4-formylphenyl)-N-phenylamino-9,9-dibutyl-fluoren-2-yl)ethynyl]-2,1,3-benzothiadiazole (**Q_1_**): A degassed mixture of **3** (330 mg, 0.55 mmol), 4,7-diethynyl-2,1,3-benzothiadiazole (**4**) (50 mg, 0.27 mmol), bis(triphenylphosphine)palladium dichloride (10 mg, 0.01 mmol), copper iodide (6 mg, 0.31 mmol) and triethylamine (0.5 mL) in toluene (5 mL) was heated at 40 °C for 48 h. The solvent was evaporated to dryness and the residue was chromatographed on silica, eluting with dichloromethane. The fractions containing the desired compound were gathered and concentrated under reduced pressure. Precipitation from pentane afforded a red orange powder (264 mg, 86%). Mp: 282 °C IR (cm-1) 2198 1693 ν_C=O_. ^1^H NMR (CDCl_3_, 300 MHz) δ 9.88 (s, 2H), 7.89 (s, 2H) 7.73 (d, J = 8.7 Hz, 4H) 7.70 (m, 10H), 7.39 (m, 4H), 7.22 (m, 10H), 7.15(d, J = 8.7 Hz, 4H) 1.97 (m, 8H), 1.13 (m, 8H), 0.75 (t, J = 7.1 Hz, 12H), 0.68 (m, 8H). ^13^C NMR (CDCl_3_, 75 MHz) δ 190.42 154.46 153.29 153.04 150.93 146.29 146.08 141.64 137.36 132.49 131.43 131.32 129.74 129.44 126.27 126.03 125.13 125.10 121.32 120.74 120.51 119.87 119.55 117.2 98.86 85.64 55.34 39.95 26.08 22.97 13.92. HRMS (FD) calcd. for C78H70N4O2S: 1126.5219. Found: 1126.5208.

#### 3.3.2. Synthesis of **Q_2_**

4,7-Bis[(7-diphenylamino-9,9-dibutyl-fluoren-2-yl)ethynyl]-2,1,3-benzothiadiazole (**Q_2_**). A degazed mixture of **2a** (342 mg, 0.60 mmol), 4,7-diethynyl-2,1,3-benzothiadiazole (**4**) (50 mg, 0.27 mmol), bis(triphenylphosphine)palladium dichloride (10 mg, 0.01 mmol), copper iodide (6 mg, 0.31 mmol) and triethylamine (0.5 mL) in toluene (5 mL) was heated at 40 °C for 48 h. The solvent was evaporated to dryness and the residue was chromatographed on silica, eluting with a 70/30 petroleum ether/dichloromethane mixture. The fractions containing the desired compound were gathered and concentrated under reduced pressure. Precipitation from pentane afforded a red orange powder (191 mg, 66%). Mp: 264 °C. ^1^H NMR (CDCl_3_, 300 MHz) δ 9.88 (s, 2H), 7.89 (s, 2H) 7.73 (d, J = 8.7 Hz, 4H) 7.70 (m, 10H), 7.39 (m, 4H), 7.22 (m, 10H), 7.15(d, J = 8.7 Hz, 4H) 1.97 (m, 8H), 1.13 (m, 8H), 0.75 (t, J = 7.1 Hz, 12H), 0.68 (m, 8H). ^13^C NMR (CDCl_3_, 75 MHz) δ 154.47 152.65 150.75 147.92 147.87 142.21 135.21 132.43 131.35 129.25 126.16 124.12 123.24 122.82 120.89 119.73 119.12 118.83 117.21 99.13 85.42 55.17 40.03 26.04 23.04 13.95. HRMS (FD) calcd. for C76H70N4S: 1070.5321. Found: 1070.5332.

#### 3.3.3. Synthesis of **Q_3_**

9,9-dibutyl-7-iodo-2-[di-(4-methoxyphenyl)amino] (**2b**): A mixture of 2,7-diiodo-9,9-dibutylfluorene (2.12 g, 4.0 mmol), di-4-methoxyphenylamine (550 mg, 2.4 mmol), potassium carbonate (3.8 g, 27.5 mmol), 18-crown-6 (40 mg, 0.1 mmol) and copper powder (180 mg, 2.8 mmol) in o-dichlorobenzene (15 mL) was degassed and heated to reflux for 24 h. When cooled down to room temperature, the suspension was diluted with dichloromethane (50 mL) and filtered on a small Celite pad. The solvents were removed by vacuum distillation and the residue was chromatographed on silica gel eluting with a 70/30 petroleum ether/dichloromethane mixture. A pale yellow solid was obtained (702 mg, 46%). Mp: 75–76 °C. ^1^H NMR (DMSO d6, 600 MHz) δ 7.73 (d, J = 1.5 Hz, 1H), 7.61 (dd, J = 7.7; 1.5 Hz, 1H), 7.60 (d, J = 8 Hz, 1H), 7.46 (d, J = 8 Hz, 1H), 6.99 (d, J= 9 Hz, 4H), 6.89 (d, J = 9 Hz, 4H), 6.88 (bs, 1H), 6.75 (dd, J=8.2; 2 Hz, 1H), 3.74 (s, 6H) 1.81 (m, 4H), 1.03 (m, 4H), 0.66 (t, J = 7.3 Hz, 6H), 0.52 (m, 4H). ^13^C NMR (DMSO–d6, 150 MHz) δ 155.87 153.01 151.35 148.89 141.01 140.79 136.08 133.12 131.83 126.37 121.47 121.34 120.22 115.66 115.31 91.99 55.72 55.13 26.23 22.82 14.26.

4,7-Bis[(7-N-(4-methoxyphenyl)-N-phenylamino-9,9-dibutyl-fluoren-2-yl)ethynyl]-2,1,3-benzothiadiazole (**Q_3_**): A degazed mixture of **2b** (250 mg, 0.41 mmol), 4,7-diethynyl-2,1,3-benzothiadiazole (**4**) (35 mg, 0.19 mmol), bis(triphenylphosphine)palladium dichloride (10 mg, 0.01 mmol), copper iodide (6 mg, 0.03 mmol) and triethylamine (0.5 mL) in toluene (5 mL) was heated at 50 °C for 24 h. The solvent was evaporated to dryness and the residue was chromatographed on silica, eluting with a 60/40 petroleum ether/dichloromethane mixture. The fractions containing the desired compound were gathered and concentrated under reduced pressure. Precipitation from pentane afforded a red orange powder (136 mg, 64%). Mp: 253 °C. ^1^H NMR (CDCl_3_, 600 MHz) δ 7.86 (s,1H) 7.65 (dd, J = 1.3; 7.8 Hz, 1H), 7.64 (d, J = 7.8 Hz, 1H), 7.61 (s,1H) 7.56 (d, J = 8.2; 1H), 7.26 (m, 2H), 7.13 (d, J = 9 Hz, 2H), 7.12 (m, 2H), 7.11 (d, J = 2.1 Hz, 1H), 7.00 (m, 2H), 6.88 (d, J = 9 Hz, 2H), 3.85 (s, 3H) 1.91 (m, 4H), 1.12 (m, 4H), 0.74 (t, J = 7.3 Hz, 6H), 0.67 (m, 4H). ^13^C NMR (CDCl_3_, 150 MHz) δ 156.14 154.46 152.57 150.66 148.27 148.17 142.34 140.84 134.46 132.43 131.33 129.14 127.03 126.12 122.97 122.04 120.78 119.49 118.97 117.57 117.18 114.76 99.19 85.37 55.53 55.11 40.04 26.02 23.04 13.96. HRMS (FD) calcd. for C78H74N4O2S: 1130.5532. Found: 1130.5550.

#### 3.3.4. Synthesis of **Q_4_**

7,7′-di-4-(trimethylsilylethynyl)-4,4′-bi-2,1,3-benzothiadiazole (**5**): A degazed mixture of 7,7′-diiodo-4,4′-bi-2,1,3-benzothiadiazole (100 mg, 0.19 mmol), trimethylsilylacetylene (80 µL, 0.56 mmol), bis(triphenylphosphine)palladium dichloride (2.6 mg, 3.8 µmol), copper iodide (0.7 mg, 3.8 µmol) and triethylamine (1 mL) in THF (4 mL) was heated at reflux for 3 h. The solvent was evaporated to dryness and the yellow residue was chromatographed on silica, eluting with a 70/30 petroleum ether/dichloromethane mixture. The fractions containing the desired compound were gathered and evaporated to dryness to yield a yellow powder (33 mg, 40%). ^1^H NMR (CDCl_3_, 300 MHz): δ 8.40 (d, J = 7.5 Hz, 2H), 7.94 (d, J = 7.5 Hz, 2H), 0.36 (s, 18H).

7,7′-Bis[(7-diphenylamino-9,9-dibutyl-fluoren-2-yl)ethynyl]—4,4′-bi-2,1,3-benzothiadiazole (**Q_4_**): A degazed mixture of **5** (41.5 mg, 0.09 mmol), **2a** (102 mg, 0.18 mmol), bis(triphenylphosphine)palladium dichloride (1.3 mg, 1.8 µmol), copper iodide (0.3 mg, 1.8 µmol), tetrabutylammonium fluoride (1 mM in THF, 0.1 mmol) and triethylamine (1 mL) in THF (4 mL) was stirred at room temperature overnight. When cooled down to room temperature, the suspension was filtered on a small Celite pad, the solvent was evaporated to dryness and the residue was chromatographed on silica, eluting with a 60/40 petroleum ether/dichloromethane mixture. The fractions containing the desired compound were gathered and evaporated to dryness. The orange powder was washed with pentane several times and dried to yield **Q_4_** (13 mg, 13%). ^1^H NMR (CDCl_3_, 600 MHz): δ 8.52 (d, J = 7.5 Hz, 2H), 8.03 (d, J = 7.5 Hz, 2H), 7.66–7.57 (m, 8H), 7.29–7.24 (d, J = 7.5 Hz, 10H), 7.05–7.00 (m, 6H), 2.00–1.82 (m, 8H), 1.15–1.04 (m, 8H), 0.75–0.66 (m, 20H). ^13^C NMR (CDCl_3_, 151 MHz): δ 155.46, 153.39, 152.78, 150.88, 147.99, 142.29, 139.31, 132.72, 132.09, 131.56, 131.34, 129.36, 126.30, 124.23, 123.38, 122.93, 121.00, 119.95, 119.23, 118.98, 117.64, 98.56, 85.58, 55.29, 40.16, 26.16, 23.16 14.07 HRMS (FD) calcd. for C82H72N6S2 + H^+^: 1206.54381, found: 1206.53492.

### 3.4. Spectroscopy

#### 3.4.1. One-Photon Spectroscopy

Photophysical properties were determined on two to three different preparations per condition (apart for dye **Q_4_** whose limited synthesised amount impeded systematic duplicate measurements). Raw absorbance spectra were measured on a Jasco V-670 UV/Vis spectrophotometer. Corrected emission spectra were measured on a Horiba Fluoromax-4 spectrofluorometer.

Molar extinction coefficients ε were obtained using Beer-Lambert law:(3)$$A\left(\lambda \right)=\varepsilon \left(\lambda \right).l.\left[C\right]$$
where A is the absorbance reading at the wavelength λ, l is the length of the optical path in the quartz cuvette and [C] is either the dye concentration or the nanoparticle concentration. Dye concentration in FONs preparations ([dye]) was determined after lyophilisation of a FONs suspension and re-dissolution of the composing dyes in an organic solvent. Nanoparticle concentration ([NP]) was determined as:(4)NP=dyeN
where N is the calculated number of dyes per nanoparticle obtained from the median dry diameter Ø_TEM_ using:(5)N=43.π.ØTEM23.ρM.NA 

With ρ, the density of the nanoparticles, assumed equal to 1 g.cm^−3^, M being the molecular weight of the dye and N_A_ the Avogadro constant (6.022 × 10^23^ mol^−1^).

When necessary, full width at half maximum values of the ICT absorption bands were determined by deconvolution of the absorption spectra using the Python SciPy library scipy.optimize.curve_fit package.

Fluorescence quantum yields were measured as the mean of values obtained against two to three standards. Measurements were repeated four to five times for FONs suspensions. Average values are shown in Table 1. Standards were chosen according to the excitation and emission range of the samples: Fluorescein (Φ_f_^FL^ = 0.9 in 0.1M NaOH), Rhodamine 6G (Φ_f_^R6G^ = 0.94 in EtOH), Cresyl Violet (Φ_f_^CV^ = 0.54 in MeOH), 4-Dicyanomethylene (Φ_f_^DCM^ = 0.44 in EtOH), Nile Red (Φ_f_^NR^ = 0.79 in DMSO). Samples and references were prepared such that the absorbance reading at the peak (of the ICT band when applicable) was below 0.1. Values were calculated using the following equation:(6)Φf=Φref.EEref.1−10−Aref1−10−A.nnref2
where Φ_f_ (Φ_ref_) is the fluorescence quantum yield of the sample (reference), E (E_ref_) is the integration of the emission spectrum of the sample (reference), A (A_ref_) is the absorbance reading at the wavelength used to excite the sample (reference) when measuring E (E_ref_), and n (n_ref_) is the refractive index of the solvent in which the sample (reference) was diluted.

#### 3.4.2. Two-Photon Excited Fluorescence Spectroscopy

2P brightness curves σ_2_.Φ_f_ = f(λ) were determined by two-photon excited fluorescence (TPEF) measurements as previously described, with the appropriate solvent-related refractive index correction [40,56]. In short, samples were excited using a Ti:Sapphire oscillator (Coherent Chameleon, Nd:YVO4, 140 fs, 80 MHz) tuned from 700 nm to 1070 nm (λ) in 10 nm increments and focused onto the sample through a 10X achromatic objective (Olympus PN, NA 0.25). Except for **Q_4_** FONs and one measurement on **Q_2_** FONs (for which detailed information of the setup can be found in our previously published work [30]), fluorescence emission was collected at 90° using a 10X achromatic objective (Olympus UPFLN, NA 0.30) and sent on a compact USB CCD spectrometer (BWTek BTC112E) through an optical fibre. Responses were measured against either Fluorescein (10^−4^ M in NaOH 0.01 M from 700 nm to 1000 nm) or Nile Red (10^−4^ M in DMSO from 900 nm to 1070 nm) used as references. The quadratic dependence of the fluorescence intensity on the excitation power was checked at all wavelengths.

#### 3.4.3. Lifetime Measurements

Time-correlated single photon counting (TCSPC) was performed on a Horiba Fluorolog spectrofluorometer using a 455 nm NanoLED as excitation source. The instrument response function was accounted for using a purely scattering sample (Ludox^®^). Fluorescence decay curves were fitted with one (dyes in solution) or three (FONs suspension) exponentials so as to determine the lifetime constants and their contributions. Measurements were repeated three times for FONs suspensions. Average values are shown in Table 1.

### 3.5. Zeta Potential Measurements

FONs zeta potentials were measured on a Horiba SZ100 Nanoparticle Analyser.

### 3.6. FONs Preparation

Dye stock solutions (1 mM in THF) were rapidly injected in freshly distilled water (1% *v*/*v*) under sonication (10 W, 3 min, room temperature). When applicable, a PAH solution (1 mM in water) was added dropwise to the FONs (1% *v*/*v*) under magnetic agitation (1200 rpm, 30 min).

### 3.7. Transmission Electron Microscopy

Dry diameters of FONs were determined by TEM. In short, undiluted samples were deposited on positively charged carbon-membrane coated copper grids. After removal of the sample drop and air drying of the grid, uranyl acetate was deposited onto the grids to contrast the objects. After removal of the contrasting agent drop and air drying of the grid, the NPs were imaged on a Hitachi H7650 electron microscope.

### 3.8. Cell Culture and Bioimaging

Cos7 monkey fibroblasts were cultured in an incubator set to 37 °C, 5% CO_2_, in DMEM (Pan Biotech) containing Foetal Bovine Serum (FBS, 10% *v*/*v*), glutamax (1% *v*/*v*) and antibiotics (penicillin/streptomycin 1% *v*/*v*). 80.000 cells were seeded in 3 cm petri dishes 48 h prior to imaging. The following day, the cells were incubated with FONs (1% or 10% *v*/*v*). Then, 30–40 min prior to imaging, the culture medium was exchanged for fresh DMEM medium equilibrated to 37 °C containing CG (30 ng/mL final concentration). Cells were then washed with and imaged in pre-warmed PBS on an upright confocal microscope (TCS SP5, Leica Microsystems) using a 25X water immersion objective (NA 0.95). CG fluorescence was excited with a 488 nm Argon laser and detected in the 500–540 nm spectral band on a photomultiplier tube (PMT). FONs fluorescence was excited with a tunable pulsed Mai-Tai 2P laser (Spectraphysics) set to 890 nm and detected in the 600–750 nm (for **Q_1_** and **Q_2_** FONs) or 630–800 nm (for **Q_3_** and **Q_4_** FONs) spectral band on another PMT. For each of the four FON types, the PMT gain was adjusted on the FONs@PAH condition so as to acquire images on the best optimised dynamic scale. It was then kept untouched for all of the following images to allow for comparison. Z-stack recordings were acquired. Maximal projection of the stacks and scaling of the resulting images was performed using Fiji/ImageJ for data representation. All images of a given condition are displayed using the same grayscale. All images are displayed with the same contrast.

## 4. Conclusions

We have studied the effect of modulating the dipolar moment of the previously described **Q_2_** quadrupolar chromophore on its spectral properties as well as on the surface properties of its FONs.

Introducing an aldehyde moiety, i.e., weakening the donor character of the triphenylamine end group of the template to yield chromophore **Q_1_**, had positive effects on its optical properties with an increase in both its molar extinction coefficient and its quantum yield. This holds true for both the chromophores in organic solvents and for the resulting FONs. Therefore, the 45% increase in ε and 25% increase in Φ_f_ make 13 nm **Q_1_** FONs twice as bright as 12 nm **Q_2_** FONs under 1P excitation while maintaining the red fluorescence emission maximum around 625 nm. Oppositely, introducing methoxy groups to the triphenylamine, and thus strengthening its donor character to yield chromophore **Q_3_**, greatly quenches fluorescence emission (−90%) to the detriment of 11 nm **Q_3_** FONs’ brightness. However, **Q_3_** nanoprecipitation also yields a subpopulation of larger, ~30 nm, FONs whose 1P brightness exceeds that of 12 nm **Q_2_** FONs by 200%. Moreover, increasing the dipolar moment of this chromophore has the anticipated advantageous effect of red shifting the emission maximum. **Q_3_** FONs, therefore, emit around 680 nm with an extended tail passed 750 nm in the NIR spectral region, making them promising bioimaging tools. In a similar fashion, **Q_1_** and **Q_3_** have opposite behaviours relative to **Q_2_** regarding their 2PA capacity. As such, the 2PA maximum of **Q_1_** FONs peaks around 750–800 nm, that of **Q_2_** around 850–900 nm, and that of **Q_3_** around 900–950 nm. Interestingly, the second strategy to increase the dipolar moment of **Q_2_**, this time by strengthening the acceptor character of the BTDA core, partly leads to similar observations. The resulting **Q_4_** chromophore indeed displays a reduced quantum yield and a red shifted emission maximum when compared to **Q_2_** but to a smaller extent than **Q_3_**. Moreover, the rotational freedom introduced by the second BTDA core appears to result in a greatly reduced 1P and 2P absorption capacity for this dye. The somewhat larger size of **Q_4_** FONs nevertheless compensates for these limited optical properties, such that their 1P and 2P brightness values are comparable to that of **Q_2_** FONs.

Finally, we confirmed the stealth character of FONs derived from this bulky hydrophobic quadrupolar chromophore template when incubated in the presence of cells. As reported previously, **Q_2_** FONs only minimally internalise in cells within 24 h, but their uptake can be triggered by coating with a positively charged polymer. Here, we further report that **Q_2_** FONs uptake does not alter cell viability and that it can be enhanced by increasing the concentration of FONs in the medium during the incubation period. **Q_1_** FONs show similar internalisation behaviour as **Q_2_** FONs, whereas **Q_3_** and **Q_4_** FONs, made from dyes bearing stronger donor or acceptor moieties, appear to be stealthier. Indeed, their uptake was negligible at both concentrations tested when incubating bare FONs even though it could be triggered by coating with PAH.

## Data Availability

The raw imaging data that enabled the generation of Figure 9 are openly available in the BioImage Archive (http://www.ebi.ac.uk/bioimage-archive accessed on 18 March 2022) under accession number S-BIAD399.

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
