# Peer review of "Stealth Luminescent Organic Nanoparticles Made from Quadrupolar Dyes for Two-Photon Bioimaging: Effect of End-Groups and Core"

_molecules, 2022, doi:10.3390/molecules27072230_

Round 1

Reviewer 1 Report

Comments:

1) Please check the usage of the abbreviations in the whole draft, like ICT in line 144, two- photon absorption in line 82, etc

2) For lines 237-240 related to the different selection rules about 1PA and 2PA, it is better to cite some typical and new references. ( Adv. Mater. 2008, 20, 4641–4678 ï¼› Angew. Chem. Int. Ed. 2009, 48, 3244 – 3266ï¼›
ACS Applied Materials & Interfaces  20211324, 28985-28995)

3) Please polish the writing languages to make this work more readable. “in between 800 and 900 nm” in line 249; “in between 720 nm and 850 nm” in line 251; and “two to three times smaller peak 2PA responses than dye Q2” in line 252;

4) Please redraw the Figure 4B and Figure 8, to make them clarity.

5) Double check the 100 MHz of the 13C NMRï¼›

6) In line 592, it is better to use “Two-photon excited fluorescence spectroscopy”

7) Please update the reference styles following the journal requirements.

Author Response

Response to Reviewer 1

We thank the rewiewer for their positive assessment of our work and their relevant suggestions to improve its strength. Below is a point by point response to their comments:

  • Please check the usage of the abbreviations in the whole draft, like ICT in line 144, two- photon absorption in line 82, etc

We thank the reviewer for noticing the inconsistencies in the use of abbreviations. We have now made our best to homogenise their use throughout the manuscript. As may be noted, we chose to maintain the non abbreviated form of “Two-photon” in section titles.

  • For lines 237-240 related to the different selection rules about 1PA and 2PA, it is better to cite some typical and new references.

In addition to the seminal paper on this topic already cited (Albota et al. Science 1998, 281, 1653-1656), we have added the works recommended by the reviewer as they indeed are relevant to illustrate our comment in Section 2.3.3 on new line 245. These include:

- Terenziani et al. Adv. Mater. 2008, 20, 4641–4678
- Pawlick et al. Angew. Chem. Int. Ed. 2009, 48, 3244 – 3266
- Feng et al. ACS App. Mat. & Int.  2021, 13, 24, 28985-28995

We also followed the reviewer’s suggestion to add more recent reviews and added the following:

- Zhang et al. Materials 2017, 10(3):223
- Klausen & Blanchard-Desce J. Photochem. Photobiol. C. 2021, 100423

  • Please polish the writing languages to make this work more readable. “in between 800 and 900 nm” in line 249; “in between 720 nm and 850 nm” in line 251; and “two to three times smaller peak 2PA responses than dye Q2” in line 252;

We have tried our best to make the manuscript as comfortable to read as we could from the first submission onwards. We apologise for any remaining inconveniences. We have tentatively rephrased the examples highlighted by the reviewer. New line 254 now reads “between 800 nm and 900 nm”; New line 256 reads “between 720 nm and 850 nm”; And new line 257 reads “Q4 shows a 2-3 times smaller peak 2PA response than dye Q2”. We are unsure how to polish the language any further as one of our authors is a native English speaker.

  • Please redraw the Figure 4B and Figure 8, to make them clarity.

We have changed the colour code for all figures so as to ensure a better contrast between the different shades in the hope that the different components are easier to discern. Regarding Panels 4B and 8A specifically, we have additionally changed the black circular markers previously used for all data points into colour-coded and shape-coded markers. As for Panel 4B, we had struggled to find an understandable depiction of FONs brightness despite the great range of values. We had settled our choice on a 3D graph to try and make the small Q3 FONs sufficiently visible but were not fully satisfied with the final representation. We now changed Panel 4B for a histogram of brightness values at each FONs’ absorption maximum. We believe this representation is clearer than the previous one while maintaining the spectral information in panel A.

  • Double check the 100 MHz of the 13C NMRï¼›

We have double-checked the 13C NMR data and changed the single-digit figures mistakenly reported for Q4 for the relevant double-digit figures.

  • In line 592, it is better to use “Two-photon excited fluorescence spectroscopy”

We have made the requested change in new line 633.

  • Please update the reference styles following the journal requirements.

We thank the reviewer for highlighting our oversight of the proper reference style. We have thus changed from ‘Vancouver’ to ‘Multidisciplinary Digital Publishing Institute’ as well as homogenised the position of the bracketed reference numbers relative to the punctuation throughout the text to better comply with the journal requirements. 

Reviewer 2 Report

The manuscript “Stealth luminescent organic nanoparticles made from quadrupolar dyes for two-photon bioimaging: Effect of end groups and core” by Blanchard-Desce et al. presents molecular-based fluorescent organic nanoparticles for two-photon absorption and their photophysical characterization, with solid and significant results. Concerning the Molecules policy, I should indicate this manuscript for publication. Nevertheless, I find that the impact of the work could considerably be augmented by some small changes in the manuscript:

- I could not find any reason to present a 3D chart without a real third axis.

- Please provide a comparison with the literature regarding the brightness results

- Regarding the TRF data. Why this characterization was performed since any discussion was presented in the manuscript? I could note quite different results from dyes and FONs. Same to the radiative decay values.

- 13C data with only one significant digit after the comma. Please revise.

- I could not find the fluorescence quantum yield standards used in this investigation.

- This submission presents supplementary material? I could not find it. Any original spectra or even TRF data (residuals, fit, decay curves) were presented.

Author Response

Response to Reviewer 2

We thank the rewiewer for their positive assessment of our work and their kind and relevant suggestions to improve its strength. Below is a point by point response to their comments:

  • I could not find any reason to present a 3D chart without a real third axis.

We agree with the reviewer that the representation of Panel 4B required improvements. We had initially struggled to find an understandable depiction of FONs brightness despite the great range of values and had settled our choice on a 3D graph to try and make the small Q3 FONs sufficiently visible. We now changed Panel 4B for a histogram of brightness values at each FONs’ absorption maximum. We hope the reviewer agrees with our belief that this representation is clearer than the previous one while maintaining the important spectral information in panel A.

  • Please provide a comparison with the literature regarding the brightness results

We have added a discussion on FONs 2P brightness values. We compare the FONs reported in this manuscript with dye-loaded silica-based nanoparticles, inorganic water soluble quantum dots and FONs described in one of our previous works, made from structurally related quadrupolar dyes. New lines 368-378 now read:

“These 2P brightness values are lower than the giant values that have been reported for dye-loaded silica-based nanoparticles (1.107 GM [48]). However, the size of these NPs is much larger (~150 nm). When 2P brightness values are considered in relation to NP size [47], our FONs (~10-20 GM/nm3) therefore compare favourably to dye-loaded silica NPs (7 GM/nm3). On the other hand, the 2P brightness of our FONs is in the same order of magnitude as that of red-emitting water-soluble quantum-dots (5.104 GM [49]) which are slightly smaller. The latter therefore display a larger 2P brightness relative to their volume (33 GM/nm3). Taken together, increasing the strength of the end-groups of our quadrupolar dyes turns out to be less efficient in terms of the ‘2P brightness vs size’ figure of merit than the modification and lengthening of the π-linker (~100-200 GM/nm3 [30]).”

  • Regarding the TRF data. Why this characterization was performed since any discussion was presented in the manuscript? I could note quite different results from dyes and FONs. Same to the radiative decay values.

We thank the reviewer for highlighting this interesting point. We had indeed hesitated about commenting on the time resolved fluorescence data and are happy to now get the chance to do so. We have thus added such a comment in section 2.4.2 which now reads (new lines 321-345):

“In addition, we note that the fluorescence decays are no longer monoexponential. Instead, FONs evidence three lifetime components (Table 1 and SI). Multicomponent decays have been observed previously in FONs made from dipolar (push-pull) or octupolar dyes [46,47]. The shortest lifetime – which is shorter than that of the corresponding molecular dye dissolved in low polarity solvents (except in the case of dye Q4) – may be tentatively ascribed to dyes located close to the FONs surface. Indeed, H-bonded water molecules promote efficient non-radiative deactivation processes via high-energy vibrations. In contrast, the longest lifetime – which is significantly longer than that of the corresponding free dye dissolved in low polarity solvents – may point to the occurrence of specific nano-arrangements of dyes within FONs (i.e. nearly parallel), which would lead to the reduction of radiative decay rates for symmetry reasons. As a whole, the reduction of the fluorescence quantum yield of the dyes induced by their confinement within pure nanoparticles can be mainly ascribed to a large decrease in the radiative rate within FONs as observed from Table 1. In contrast, we note that the non-radiative decay rates are reduced in comparison to those of the dyes dissolved in medium polarity solvents, and similar to those of the dyes in a low polarity solvent (except for dye Q2). Along this line, we stress that the fluorescence quantum yield of FONs in water (εr = 80) is significantly larger than that of their corresponding dye in a polar solvent such as acetonitrile (εr = 31), where all dyes are non-fluorescent. This clearly illustrates that FONs made from quadrupolar dyes can be considered as leading to aggregation caused quenching (ACQ) when compared to dyes dissolved in THF, but to aggregation induced emission (AIE) when compared to dyes dissolved in a polar solvent. Despite the small FONs fluorescence quantum yield values in water, all FONs show large brightness values thanks to the high number of confined dyes per FON and to the large absorption coefficients of the quadrupolar dyes (Table 3).”

  • 13C data with only one significant digit after the comma. Please revise.

We have double-checked the 13C NMR data and changed the single-digit figures mistakenly reported for Q4 for the relevant double-digit figures.

  • I could not find the fluorescence quantum yield standards used in this investigation.

We apologise for this oversight and have added the standards used in the materials and methods section in new lines 618-621. Namely, Fluorescein, Rhodamine 6G, Cresyl Violet, 4-Dicyanomethylene or Nile Red have been used depending on the compound measured.

  • This submission presents supplementary material? I could not find it. Any original spectra or even TRF data (residuals, fit, decay curves) were presented.

We had indeed not submitted any supplementary material with this work. On the reviewer’s kind suggestion, we have now included one to illustrate the TRF curves, fits and residuals.